# Comment on Jun, S.Y.; et al. “Tumor Budding and Poorly Differentiated Clusters in Small Intestinal Adenocarcinoma” *Cancers* 2020, *12*, 2199

**DOI:** 10.3390/cancers12102982

**Published:** 2020-10-15

**Authors:** Paolo Giuffrida, Giovanni Arpa, Alessandro Vanoli, Antonio Di Sabatino

**Affiliations:** 1First Department of Internal Medicine, University of Pavia and Fondazione IRCCS Policlinico San Matteo, 27100 Pavia, Italy; paolo.giuffrida01@universitadipavia.it; 2Anatomic Pathology Unit, Department of Molecular Medicine, University of Pavia and Fondazione IRCCS Policlinico San Matteo, 27100 Pavia, Italy; giovanni.arpa90@gmail.com (G.A.); alessandro.vanoli@unipv.it (A.V.)

We read with interest the paper by Jun S.Y. et al. [1] dealing with the tumor budding (Tb) and poorly differentiated clusters (PDCs) in small bowel adenocarcinoma (SBA). However, in contrast to what the authors stated, this is not the first study investigating the clinicopathologic significance of TB and PDC and their prognostic values in SBAs. Indeed, we have recently assessed Tb and PDCs in non-ampullary SBAs in a cohort of 47 Crohn’s disease patients [2]. Moreover, Tb was studied in ampullary adenocarcinomas, in which high-grade Tb was found an independent predictor of overall survival [3]. Jun S.Y. et al. [1] confirmed our findings, showing that high-grade Tb or PDCs are associated with an aggressive behavior and, thus, with a worse patient outcome in SBA [2]. Therefore, the main novelty of this study was to identify the invasive front markers Tb and PDCs as prognostic indicators in sporadic SBAs [1], whereas we obtained the same results in Crohn’s disease-associated SBAs [2]. Finally, we do acknowledge that Jun S.Y. et al. [1] are the first to analyze Tb and PDCs in the intratumoral area of SBAs. 

As survival rate is similarly low in patients with Crohn’s disease-associated SBA and those with sporadic SBA [4,5,6], it is clinically relevant to find prognostic markers for this neoplasm, in particular in these two etiologic groups. Briefly, these findings should encourage pathologists to describe Tb and PDCs at SBA diagnosis in both Crohn’s disease patients and sporadic cases in order to separate highly malignant cancers from less aggressive SBAs. Indeed, Tb and PDCs may represent pivotal histological features together with mismatch repair status, histotype, and pT [7] to identify stage II SBAs needing adjuvant chemotherapy or further surgery. As well, they have been demonstrated in colon adenocarcinoma [8].

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
