# Peer review of "Comment on Jun, S.Y.; et al. “Tumor Budding and Poorly Differentiated Clusters in Small Intestinal Adenocarcinoma” Cancers 2020, 12, 2199"

_cancers, 2020, doi:10.3390/cancers12102982_

Round 1

Reviewer 1 Report

I kindly disagree with the Authors of this Comment since inflammatory disease-associated and sporadic small bowel adenocarcinomas are genetically and clinically different.

Reviewer 2 Report

This is an odd situation.

The comment is correct. There is an online publication dating from
August 16, 2019 but it was not published formally by the journal until
the March 2020 issue.

This probably is what accounts for the missing citation in the
literature review of the paper published in Cancer.

You are going to need the opinion of a pathologist that specializes in
these types of tumors to determine if this comment should be published,
because this will come down to the significance of the difference
between sporadic versus Crohn's disease association for SBA.
Unfortunately, I am a synthetic chemist, and I am not the right referee
to make that determination. If I were the author's of this comment, I
would not have sent this to you.